# Tracking the Morphological Trends in *Apodemus flavicollis*: Evidence from a Five-Decade Study

**DOI:** 10.3390/life15020322

**Published:** 2025-02-19

**Authors:** Linas Balčiauskas, Laima Balčiauskienė

**Affiliations:** Nature Research Centre, Akademijos 2, 08412 Vilnius, Lithuania; laima.balciauskiene@gamtc.lt

**Keywords:** yellow-necked mouse, mid-latitude, morphometric traits, age groups, size shrinking

## Abstract

We examined long-term trends in the yellow-necked mouse (*Apodemus flavicollis*) in Lithuania using data collected from 1980 to 2024. Over 10,000 individuals were captured and 5666 were necropsied to measure standard morphometric traits, including body mass, length, and appendage dimensions. Temporal trends revealed an increase in the proportion of *A. flavicollis* within small mammal communities, from 6.5% in the 1980s to 28.2% in the 2020s, particularly in forest and grassland habitats. Sexual dimorphism was confirmed, with males generally larger than females in all traits, although age influenced the degree of size difference. Morphometric traits showed a consistent increase from the 1980s to the 2010s, followed by significant declines in body mass, body length, and ear length in the 2020s. Declines were most pronounced in males across all age groups. These findings are consistent with our previous findings in the field vole (*Microtus agrestis*) from the Baltic region, and the global patterns of body size reduction in small mammals due to climate warming and habitat change. This study highlights the importance of integrating sex- and age-based analyses to understand the ecological and evolutionary responses of mammals to environmental pressures. Future research is essential to explore the broader implications of these trends for ecosystem dynamics and species survival.

## 1. Introduction

Historically, mammalian body size has increased up to the limits imposed by resource availability [1]. Mammalian body size is measurable and therefore suitable as an indicator of species ecology and life history [2]. Due to the importance of body size in the physiology of an individual—defined as “production per unit of body mass is necessarily slower in larger organisms than in smaller ones”—body size parameters are important for understanding changes in the lifestyles of different mammalian species [3]. Climate change has been implicated in driving mammalian shifts, from past extinctions to modern impacts [4], with evidence linking these to human-altered landscapes [5].

Ecologists have already accumulated a considerable body of knowledge documenting the various changes in Europe’s mammals over the past 200 years [6]. Studies cover insectivores [7], bats [8,9,10], rodents [11], carnivores [12,13,14,15,16], and ungulates [17]. These studies indicate both increases (e.g., bats [8,9,10], foxes [12], martens [15], badgers [12]) and decreases (e.g., shrews [7], voles [11], martens [14], moose [17]) in mammalian body size, depending on species and region.

Within a species, however, there may be mixed patterns of size change. For example, during five decades in Sweden, Eurasian otters (*Lutra lutra*) showed significant increases in skull size and body mass, and skull size but not body mass; this is negatively related to latitude, in contrast with Bergmann’s rule [13]. The authors attributed the increase in size to a combination of rising ambient temperature and increased food availability due to longer ice-free periods. Another study in Sweden, covering three decades for weasels (*Mustela nivalis*) and nearly eight decades for stoats (*M. erminea*), found that male skull size was positively correlated with ambient temperature and net primary productivity, likely due to energy conservation or improved food availability, while females showed no such relationship, suggesting different selection pressures between the sexes [16]. In addition to climate change, habitat fragmentation has been implicated as a selective force driving mammal body length in the Nordic environment [18].

In the perspective paper published by J. Sheridan and D. Bickford in 2011, size shrinkage was considered characteristic of both aquatic and terrestrial ectotherms, along with higher taxa, birds, and mammals [19], and the authors expected this conclusion to be supported by further studies. However, studies of carnivores showed that temporal changes were significant in only 6 out of 52 populations studied [20]. The rate of change in body length was generally lowest in medium-sized mammals and increased with both smaller and larger body mass. Therefore, small mammals may be a good reference group for such analyses. However, contrary with other authors, some concluded that most small mammals have increased [18].

Several analyses indicate that the body size of European small mammals has been decreasing over the last decades. Shrews may not be the best example for temporal analysis due to seasonal size changes (Dehnel’s phenomenon), which must be taken into account in the analyses [21]. Nevertheless, during a 50-year study in Poland, the skull size of the common shrew (*Sorex araneus*) decreased in relation to warming and decreasing soil moisture, which made earthworms, the main food of shrews, less available [7]. Despite a male bias in skull size, the degree of decline did not differ between sexes. In a long-term study of several soricid species in Slovakia, the geographic pattern was considered to have a stronger effect than the temporal effect [22].

Similar trends of body and skull shrinkage were observed in the field vole (*Microtus agrestis*), where body and skull dimensions in recent decades showed a pronounced decrease in Estonia and a less pronounced trend in Lithuania [11]. The other example of decline in body mass and dimensions in the pine vole (*Microtus subterraneus*) and the yellow-necked mouse (*Sylvaemus flavicollis*) comes from Ukraine, where this trend was influenced by habitat changes under anthropogenic conditions [23]. We use the name *Apodemus flavicollis* for the same species, as found in the Global Biodiversity Information Facility database [24].

When analyzing sexually dimorphic species of small mammals, samples must be stratified by sex [25] and age [26]. Dimorphism in small mammals is best expressed in voles [27], such as the root vole (*Alexandromys oeconomus*) or the Tatra vole (*Microtus tatricus*) [28,29], while, in the bank vole (*Clethrionomys glareolus*), it is not uniform [30,31,32]. As a reference, in Lithuania, 10 out of 14 species of small mammals have been shown to be dimorphic, classified as male-biased or female-biased, while 4 other species, mainly shrews, have been classified as monomorphic [25]. As for *A. flavicollis*, the dimorphism of this species is well known [33,34,35,36,37] but has been shown to be very variable [25,38]. In conclusion, most of the known long-term trends indicate a decrease in body size of European small mammals (mice, voles, shrews), driven by rising temperatures and habitat changes, with notable differences depending on sex and age.

The only case study of temporal changes in *A. flavicollis* found a decreasing trend in body size, especially in females [23]. In contrast, a 100-year study of three rodent species, *A. flavicollis*, *A. sylvaticus*, and *M. agrestis*, in Denmark showed no changes in body size depending on animal sex, latitude, or year [39].

Thus, while anthropogenic effects and some sex-specific size differences have been identified in *A. flavicollis*, there is still a gap in the longitudinal data that rigorously analyze body size variation over time in relation to environmental changes. Such analyses based on simple morphometric parameters could reveal which species are unable to adapt to their changing environment [19]. As a response to encourage scientists in other countries to perform similar analyses of their national data [23], we analyzed changes in body size of *A. flavicollis* in Lithuania. The aim of the study was to characterize sexual size dimorphism and age-related dimorphism in yellow-necked mice (*Apodemus flavicollis*) in the country, based on standard morphological traits, and to assess temporal changes in these traits since the 1980s.

## 2. Materials and Methods

### 2.1. Sampling and Sample Size

Small mammals were trapped in Lithuania between 1980 and 2024 (we have no retrospective data on measurements from earlier trapping). The standard method of snap trap lines was used (25 traps 5 m apart, arranged in a line). More details on trapping are presented in [25]. As a standard bait, we used cubes of black bread crust, about 1 cm in size, soaked in crude sunflower oil. As a standard method, trap lines are designed to collect a representative sample of the species present in the community rather than exhaustively capture all animals, as this could cause significant disturbance to the ecosystem. Trapping was standardized across the years to ensure compatibility and minimize bias. Our adherence to standardized protocols allowed us to retrospectively estimate the relative proportions of different species within the sample rather than absolute population sizes.

A total of 54,116 small mammal individuals, including 10,027 *A. flavicollis*, were captured during the study period. Of these, 29,203 small mammals, including 5666 *A. flavicollis*, were dissected and measured. The sample composition by decade is shown in Table 1, indicating the age structure of dissected *A. flavicollis*.

Small mammals were kept refrigerated without freezing if measured and dissected the same day after capture, or were frozen one at a time in plastic bags and stored in them until they were transferred to the laboratory.

### 2.2. Morphometric Traits of Small Mammals

We collected five standard measurements—body mass (Q), body length (L), tail length (C), hind foot length (P), and ear length (A)—according to [40,41]. Q was measured to the nearest 0.1 g using spring, weighing, or electronic scales; L, C, P, and A were measured to the nearest 0.1 mm using mechanical or electronic calipers in a standard manner [41]. L was measured from the tip of the nose to the posterior margin of the anus in the supine position; C, from the posterior margin of the anus to the tip of the tail, excluding hair; P of the right foot, from the dorsum of the heel to the tip of the middle toe, excluding the claw; and A, from the notch at the base of the right ear to the distal edge, excluding hair. The first author measured over 80% of the small mammals in the sample throughout the study period, minimizing potential bias according to [42]. Uterine and embryo weights of pregnant females were subtracted from body weight.

Small mammals were measured post-mortem before necropsy. Their body states were primary flaccidity (necropsy on the same day of capture) and secondary flaccidity (frozen and fully thawed) according to [43]. Necropsies under primary flaccidity were mostly performed in the 1980s, while, after 1990, necropsies were mostly performed on refrozen individuals.

Three age groups of *A. flavicollis* (juveniles, subadults, and adults) were identified at necropsy, based on development of sexual organs, thymus atrophy, and occasionally body mass. Body mass alone was considered unreliable due to variability between age groups. Juveniles were characterized as non-breeding individuals with a well-developed thymus, filiform uterus (females), or abdominal testes (males). Subadults were characterized as non-breeding individuals with partially involuted thymus glands and inactive genitalia. Adults were characterized as individuals with an atrophied thymus and reproductive activity (e.g., lactation, pregnancy, corpora lutea, placental scars in females; scrotal testes and developed seminal vesicles in males) or post-reproductive signs [25].

### 2.3. Data Analyses

Differences in *A. flavicollis* proportions in small mammal communities across decades were tested using the χ^2^ statistic.

For most morphometric traits across age and sex categories, normality of the distribution was not confirmed by Kolmogorov–Smirnov or Shapiro–Wilk tests [44].

We performed a general linear model (GLM) analysis to evaluate the significance of different factors. In this analysis, the dependent variables were Q, L, C, P, and A, while the independent variables included a categorical factor (decade) and covariates (sex and age of individuals). The aim was to determine the significance of each factor and to assess whether temporal variation occurred in the dependent variables. Effect size was assessed using partial eta-squared (*η*^2^).

Having determined that age and sex had a strong influence on the dependent variables, we analyzed the temporal changes in Q, L, C, P, and A while controlling for animal age and sex. This was performed using ANOVA with “decade” as the grouping factor. We relied on the ability to use ANOVA when the data were continuous and the samples were randomly drawn from the species populations [45], even if normality was not confirmed. To evaluate whether the variation in each dependent variable was significant, we used Wilks’ Lambda as the test statistic. Post hoc comparisons were performed using the Bonferroni correction, which adjusts *p*-values or significance thresholds to account for the number of comparisons. This method was chosen because it does not make any specific distributional assumptions, making it less dependent on the assumptions of ANOVA.

Calculations were performed in Statistica for Windows version 6.0 (StatSoft, Inc., Tulsa, OK, USA) [46]; normality tests and chi-square tests in PAST version 4.13 (Museum of Paleontology, Oslo College, Oslo, Norway) [44]. The minimum confidence level was set as *p* < 0.05.

## 3. Results

### 3.1. Proportions of Apodemus flavicollis in Small Mammal Community

Our data showed a constant overall increase in the proportion of *A. flavicollis* across decades, from 6.5% in the 1980s to 28.2% in the 2020s (Table 2). This increase was characteristic for forest habitats. Agricultural habitats (in the 2010s and 2020s also including commercial orchards and berry plantations) were always important. In the 2020s, the importance of meadows and agricultural areas increased.

In forest habitats, proportions of *A. flavicollis* were characterized by a steady and significant increase over time, with an increasing trend between the 2010s and 2020s that did not reach the significance threshold (χ^2^ = 3.19, *p* = 0.074). In grasslands, the proportion of *A. flavicollis* was stable and relatively low from the 1980s to the 2010s, with small non-significant changes, until a significant and sharp increase in the 2020s. In the wetlands, there was a significant increase in proportions from the 1980s to the 2000s, followed by a significant decrease in the 2010s. In agricultural areas, there were relatively stable proportions in the first four decades, followed by a significant increase in the 2020s.

Thus, a significant upward trend over time, with substantial increases in the proportion of *A. flavicollis* in small mammal communities in the 2010s and 2020s, was mostly due to forest and grassland habitats becoming increasingly important for the species.

### 3.2. Sex- and Age-Based Size Dimorphism in Apodemus flavicollis

Male-biased sexual size dimorphism is characteristic of *A. flavicollis*, with some corrections in different age groups (Figure 1). In adult animals, males are larger in all traits; differences are significant at *p* < 0.0001. Expressed as percentages, adult male body mass exceeds female body mass by 11.2%, body length by 3.8%, tail length by 2.2%, hind foot length by 3.4%, and ear length by 2.9%. In subadult *A. flavicollis*, males are 8.9% larger than females in body mass, 2.4% larger in body length, 1.8% larger in hind foot length (all *p* < 0.001), 1.4% larger in tail length (*p* < 0.005), and similar in ear length. In juvenile *A. flavicollis*, males are larger by 0.8% in hind foot length (*p* < 0.05), but females are larger in body length and tail length (1.5% and 1.9%, respectively, *p* < 0.025), while body mass and ear length do not differ (*p* > 0.05).

Thus, male-biased sexual size dimorphism in *A. flavicollis* is evident across age groups, with males being significantly larger in most traits in adults and subadults, while juveniles show minimal dimorphism, including cases where females exceed males in certain traits.

### 3.3. Overlap of the Morphometric Traits in Age Groups

The comparison of morphological traits (body mass, body length, tail length, hind foot length, and ear length) across different life stages (juvenile, subadult, and adult) in males and females of *A. flavicollis* shows the considerable overlap in trait ranges between age groups, especially for measurements such as body mass, body length, and tail length (Table 3). This overlap suggests a gradual progression of morphological traits with growth and development rather than discrete shifts between age groups.

### 3.4. Temporal Changes in Morphometric Traits Depending on Sex and Age

The analysis of changes in body mass over the decades shows a consistent decrease in body mass in the 2020s in all age groups in both sexes, following the upward trend of the previous decades (Figure 2). The decrease is better expressed in males of all age groups: juveniles (F_4,577_ = 3.8, *p* < 0.005), subadults (F_4,787_ = 15.6, *p* < 0.0001), and adults (F_4,1574_ = 26.1, *p* < 0.0001). In juvenile females, body mass increased after the 1980s, with no difference between the 2010s and 2020s. In subadult and adult females, after the upward trend from the 1980s to 2010s, the decrease in body mass in the 2020s was less pronounced than in males but still highly significant (F_4,846_ = 12.8, *p* < 0.0001 and F_4,1029_ = 4.6, *p* < 0.002, respectively).

Body length showed a decreasing trend that started in the earlier decades than the decrease in body weight. In non-juvenile age classes of *A. flavicollis*, the largest reductions were observed after the 2000s (1990s in the case of adult females) and continued into the 2020s (Figure 3). Juvenile females were characterized by a stable body length throughout the study period (F_4,699_ = 2.1, *p* = 0.07). Juvenile males showed a decrease in body length from the 2000s onwards (F_4,570_ = 3.8, *p* < 0.005), though the decrease in the 2020s compared with the 2010s was not significant. The decrease in body length in subadult males and females (F_4,779_ = 5.5 and F_4,844_ = 8.1) and in adults (F_4,1564_ = 14.8 and F_4,1022_ = 13.9) was highly significant in all cases, with *p* < 0.0001.

Thus, the decline in body length began earlier than the decline in body mass and was evident in all age groups (juveniles, subadults, and adults) and in both sexes.

Changes in tail length over the study period showed variability, with juveniles showing greater variability than older groups (Figure 4). Variation in the trait in juvenile males and females was significant (F_4,354_ = 3.2, *p* < 0.05, and F_4,467_ = 4.9, *p* < 0.001, respectively), with tail length decreasing in the 2020s in both sexes. In subadult males, tail length variation was significant across decades (F_4,446_ = 5.5, *p* < 0.001), with a notable decrease in the 2020s. In subadult females, tail length remained stable (F_4,489_ = 1.2, *p* = 0.30), with a non-significant increase in the 2020s. In adult males, tail length decreased markedly in the 2020s, with fluctuations across decades being significant (F_4,960_ = 3.8, *p* < 0.005), as in females (F_4,659_ = 2.7, *p* < 0.05), but a decrease in the 2020s was not expressed.

Hind foot length in juvenile males increased from the 1980s to 2000s and stayed at that level (F_4,371_ = 5.7, *p* < 0.001); in juvenile females, we did not observe significant changes (F_4,480_ = 0.3, *p* = 0.85). In subadult males, the decrease in hind foot length after the 1990s (Figure 5) was not significant (F_4,462_ = 0.4, *p* = 0.80), and in subadult females the changes were reliable (F_4,494_ = 4.9, p < 0.001) but not directional. In adult *A. flavicollis* males (F_4,1000_ = 23.5, *p* < 0.0001) and females (F_4,689_ = 12.3, *p* < 0.0001), hind foot length fluctuated widely and it cannot be said that there has been a downward trend over the last decade. On the contrary, in subadult and adult females, hind foot length increased in the 2020s compared with the 2010s (Figure 5).

The decreasing trend in ear length of *A. flavicollis* was observed from the 1990s to the 2020s across all age classes and sexes. This trend was particularly pronounced after the 2010s (Figure 6). Due to the high variability of ear length in juveniles, the decrease in the 2020s was pronounced but not significant in males (F_4,360_ = 0.94, *p* = 0.44) and weakly significant in females (F_4,472_ = 2.5, *p* < 0.05). In subadult males and females, the decreasing trend was significant (F_4,453_ = 5.3, *p* < 0.001 and F_4,487_ = 3.0, *p* < 0.02, respectively), with the decrease in the 2020s compared with the 2010s being > 5%. The decreasing trend in ear length after the 1990s was near significant in adult males (F_4,986_ = 2.23, *p* = 0.06) but was significant in adult females (F4,679 = 6.6, *p* < 0.0001).

### 3.5. General Trends of Morphometric Trait Changes in Apodemus flavicollis

After the GLM modeling of temporal changes in morphometric traits, we found that the effect of decade was highly significant (Wilks λ = 0.77, *p* < 0.0001, *η*^2^ = 0.066), although smaller than that of individual age (λ = 0.42, *p* < 0.0001, *η*^2^ = 0.578) and sex (λ = 0.86, *p* < 0.0001, *η*^2^ = 0.141). Based on this, we evaluated the general temporal patterns calculated for the covariates of sex and age (Figure 7).

In general, increases occurred from the 1980s to the 1990s in hind foot length and ear length, from the 1980s to the 2000s in body length, and from the 1980s to the 2010s in tail length (Figure 7), all of which were significant (post hoc, *p* < 0.05 or higher). After the 2010s, significant decreases were observed in body weight (post hoc, *p* < 0.0001), body length (*p* < 0.05), tail length (*p* > 0.001), and ear length (*p* < 0.0001). There were no changes in hind foot length in the last two decades, but values in the 2010s–2020s were significantly lower than in the 1990s–2000s (post hoc, *p* < 0.001).

## 4. Discussion

Our analyses revealed three characteristics of *A. flavicollis* populations in Lithuania during the period from the 1980s to the present. The first is the increase in the proportion of *A. favicollis* in small mammal communities, especially in forest and meadow habitats, and the constant presence of this species in agricultural habitats. The second is the body dimorphism of this species, which favors males. The third is the reduction in body mass, body length, and ear length over time, especially in the 2020s.

### 4.1. Proportion of Apodemus flavicollis in Small Mammal Communities of Different Countries

We found a fourfold increase in *A. flavicollis* proportion, from 6.5% in the 1980s to 28.2% in the 2020s, with the most pronounced increase in forest habitats. This increase corresponds to the decreasing abundance and proportion of *C. glareolus* in the country since the 1990s [47].

An increase in *A. flavicollis* proportion was documented in the Czech Republic, from 42% in 1968–1972 to 62% in 2002–2006 under conditions of decreasing moisture in floodplain forests [48]. In mature beech and young spruce forests of the Czech Republic, the average *A. flavicollis* proportion in 1997–2015 was 42.2%, and, during this time, the average species abundance doubled [49]. An even stronger increase was found in Poland, from 2.6% in 1981 to 24.5% in 2008, as a result of the fallowing of arable fields [50], this figure being similar as we observed in Lithuania (see Table 2). Polish researchers noticed an increase in *A. flavicollis* abundances even earlier, reporting an increase in total species abundance [51]. In the same country, however, temporal trends may differ. Temporal trends in the presence and relative abundance of *A. flavicollis* in three habitat types—lakeshore, forest, and farmland—from the 1970s to the 2000s are available from Poland [52]. In the lakeshore habitat, the authors showed a fluctuating occupancy over time, with peaks in the 2000s at 31–50%. In the forest habitat, *A. flavicollis* proportions peaked in the 1990s at 31–50% and over 50%, with a decline in the 2000s. In arable land, the proportion of the species increased after the 1980s, but data for this habitat are fragmented [52].

Temporally stable proportions of *A. flavicollis* have been described in Latvia [53] and Sweden, where the average abundance from 1961 to 1988 was about one individual per 100 trap nights, fluctuating insignificantly with no expressed trend [54]. Decreases in *A. flavicollis* abundance have also been described. In the woodland habitat of Woodchester Park, UK, the proportion was 45.7% in the 1960s, 38.2% in the 1970s, and further decreased to 17.7% in the 1980s [55]. In Slovakia, the proportion of *A. flavicollis* decreased from 71.2% in 1987–1988 to 57.4% in 1992–1994, and further to 38.6% in 2002–2004, all related to the same area near Košice [56].

In conclusion, the proportion of *Apodemus flavicollis* in small mammal communities has generally increased over time, particularly in forest habitats in different countries, but temporal trends vary between regions and habitats, with notable declines in some areas such as the UK and Slovakia. However, the above-cited authors offer no unifying explanation for the observed changes, as the trends appear to be influenced by diverse and region-specific factors.

### 4.2. Is Apodemus flavicollis Dimorphic?

The first data for Lithuania reported *A. flavicollis* to be monomorphic in body mass and length, as “adult males and females are of the same size” [32]. The other studies, however, do not confirm this and indicate a male-biased dimorphism in both body mass and length.

The strongest male bias is reported as male body weight being 17.7% greater than female body weight [37], followed by biases such as 16.6% [33], both in Poland and in different decades. Moderate male bias, 6.6–8.6%, concerns Slovakia [36,57], with minimum 1.5% again in Poland [34]. Male bias in body length of *A. flavicollis* is less expressed; males are longer than females by 4.4% in Poland [33], by 3.3% in Great Britain [58], and by 1.8–2.0% in Slovakia [36].

Therefore, our data showing a male bias in body size, ranging from 2.2% in tail length to 11.2% in body mass of adult *A. flavicollis* individuals, are consistent with the published data.

### 4.3. Overlap of the Morphometric Traits in Age Groups of Apodemus flavicollis

Yellow-necked mice, like other *Apodemus* species, continue postnatal growth throughout the entire life span [59]; therefore, the overlap of morphometric traits in different age groups is of interest for understanding the life history of the species. One of the questions is the minimum body mass in mature individuals. In our study, the minimum body mass of adult females was 15.0 g, that of males was 18.1 g.

The studies of the 1940s and 1950s considered the minimum body weight for maturity in females as 22–30 g; in the southernmost parts of the range, 16.5–17 g; in males, the minimum body weight was reported as 30–34 g, and 28 g in the south, referring to sources cited after [60]. In Slovakia, data from the 2000s indicate that the minimum body mass of adult females was 23 g, that of males 24 g, while the body length was 87 mm and 88 mm, respectively [36]. The same study showed a very high overlap in morphometric traits between adult and subadult age categories. The most overlapping traits were hind foot length and ear length, so our study is fully compatible. Other comparable data are not available.

### 4.4. Temporal Changes in Morphometric Traits in Apodemus flavicollis

As a general pattern for the species, we found an increase in morphometric traits in *A. flavicollis* from the 1980s to the 1990s (hind foot and ear length), 2000s (body length), and 2010s (tail length). After the 2010s, significant declines were observed in all traits except hind foot length, for which the decline started in the 2000s. At a detailed level, these temporal changes had differences according to the sex and age of the animals (see Figure 2, Figure 3, Figure 4, Figure 5 and Figure 6).

The only study dealing with long-term changes of morphometric characteristics of *A. flavicollis* concerns Ukraine. It shows a decrease in body length from the 1970s to the 2000s by 17.1%, decrease in body height by 10.1%, and decrease in fatness index by 16.7% [23]. The author pointed to the decrease in other traits, such as tail length, hind foot length, and ear length, but no values were given, only the mention that the decrease in body mass was higher. It was also shown that the decrease was greater in females and in adult animals. This was also characteristic for pine voles (*Microtus subterraneus*) [23]. In short, our results are similar.

The idea that small mammals have generally increased in length while large mammals have decreased during 200 years in Denmark [6] is not clear; the data presented by the authors in the paper are unbelievable. For example, recent body length was reported as 15.6 cm in *C. glareolus*, 11.6 cm in *S. araneus*, 14.06 cm in *M. agrestis*, 12.9 cm (!) in harvest mouse (*Micromys minutus*), and 18.7 cm in striped field mouse (*Apodemus agrarius*), while average values from other sources were reported as 10–11 cm, 4.8–8.0 cm, 11.5–13 cm, 5.5–7.5 cm, and 9.4–11.6 cm, respectively [61,62]. This difference explains their presumably wrong conclusion on size increase [6]. Another study, based on the same data from Denmark, reported that year, latitude, and sex were not related to body size of *M. agrestis, A. flavicollis,* and wood mouse (*Apodemus sylvaticus*) [39], which is not consistent with our results and those of other authors.

On the contrary, an analysis of body size changes in North American small mammals during the last century indicated that these species were shrinking. Size reduction was found in 23 species of sciurids, cricetids, heteromyids, geomyids, zapodids, and soricids related to body mass, body length, hind foot length, and ear length, while tail length increased [63]. Tropical mammals have all shown a reduced size or growth rate during drought years, under experimental desiccation conditions, or along a decreasing precipitation gradient [19].

### 4.5. Drivers and Importance of Temporal Body Size Changes

Climate warming has been implicated as a major driver of body size reduction in animals [4,5]. However, “climate change exerts a profound yet complex influence on mammals, emphasizing the importance of continued research to disentangle phenotypic plasticity from evolutionary adaptations”, i.e., changes may be related to the phenotype or genotype of the studied species [64]. It is possible that female rodent dimensions are evolutionarily hardwired to remain stable, whereas males may be more sensitive to changes in body size due to their life history characteristics, including greater energetic demands and behavioral exposure to environmental risks. For example, studies of prairie voles (*Microtus ochrogaster*) identified ontogenetic growth differences as drivers of male size shifts, mediated by density-dependent environmental factors [65]. Thus, while some studies suggest a pronounced decline in male body size, the mechanisms underlying these trends are not well understood.

However, there are other less understood drivers of mammalian size change. Urbanization (as a proxy for human population density) has been shown to influence small mammal body size in North America [66]. These authors conclude that both urbanization and temperature often lead to larger individuals as resource availability increases. In contrast, other authors estimate that, by 2100, the reduction in body size due to all anthropogenic drivers could be in the range of 10–21% [63]. The importance of habitat change as a driver of small mammal body size reduction is also supported by [23].

Changes in body size may challenge the survival of species during climatic extremes [19]. Including predator–prey relationships in insectivores, dietary changes in other small mammal groups and morphological shifts in their bodies could result in cascading effects on the structure and functioning of terrestrial communities across the globe [63]. On the other hand, small mammals are a resilient group that can dampen the effects of environmental variability through microhabitat use and flexible activity patterns [67]. Therefore, comprehensive longitudinal studies are essential to better understand the long-term effects of environmental change on body size dynamics in broader geographic contexts, as was shown for *A. agrarius* [68].

As for Lithuania, its climate has undergone significant changes from the 1990s to the 2020s, consistent with broader regional and global climate trends. The country has experienced warming and shifts in precipitation patterns. During these three decades, average annual air temperatures increased by +1.0 °C, with the largest increases observed in November and December (+3.4 °C and +3.3 °C, respectively). Annual precipitation decreased slightly, from 686 mm to 652 mm, with notable changes in the summer months, increasing by 10.9 mm and 22.9 mm in July and August, respectively, but decreasing in spring and fall. Warmer winters led to earlier snowmelt [69]. In recent years, 2021–2023, these changes remained in power [70], confirming forecasts for the future. It is expected that the average annual temperature will increase by 1.1–1.4 °C by 2035 and by 1.5–5.1 °C by 2100, with the greatest changes expected in cold season temperatures [71]. In view of the above, it is very likely that the increase in the proportion of *A. flavicollis* in the community and the decrease in morphometric indices, which have been particularly pronounced in the last decade, are related to the expression of climate change in Lithuania.

## 5. Conclusions

Our data and analyses revealed three long-term trends in *A. flavicollis* populations as follows:Increasing proportion of the species in small mammal communities, especially in forest and grassland habitats.Male-biased body size dimorphism, most pronounced in adult individuals.Decline in body mass, body length, tail length, and ear length over time, especially in the 2020s.

## Figures and Tables

**Figure 1 life-15-00322-f001:**
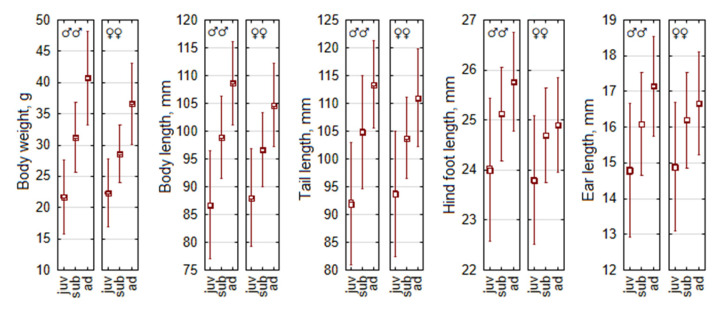
Statistics of five main morphometric traits in *Apodemus flavicollis* by sex and age. Central tendency and variability are represented by mean (square), standard error (box), and standard deviation (whiskers). Differences between age groups on all dimensions are reliable for both sexes (*p* < 0.001).

**Figure 2 life-15-00322-f002:**
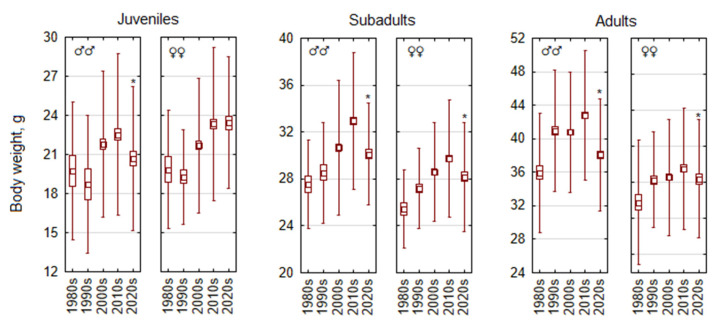
Temporal changes in body mass of *Apodemus flavicollis* from 1980 to 2024 depending on sex and age. Central tendency and variability are represented by mean (central square), standard error (box), and standard deviation (whiskers). Significant trait changes in the 2020s compared with 2010s marked with an asterisk.

**Figure 3 life-15-00322-f003:**
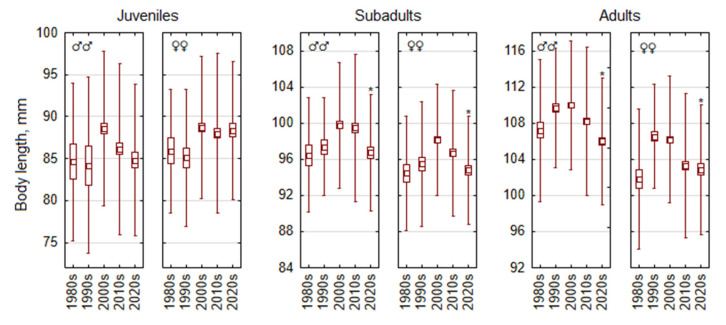
Temporal changes in body length of *Apodemus flavicollis* from 1980 to 2024 depending on sex and age. Central tendency and variability are represented by mean (central square), standard error (box), and standard deviation (whiskers). Significant trait changes in the 2020s compared with the 2010s are marked with an asterisk.

**Figure 4 life-15-00322-f004:**
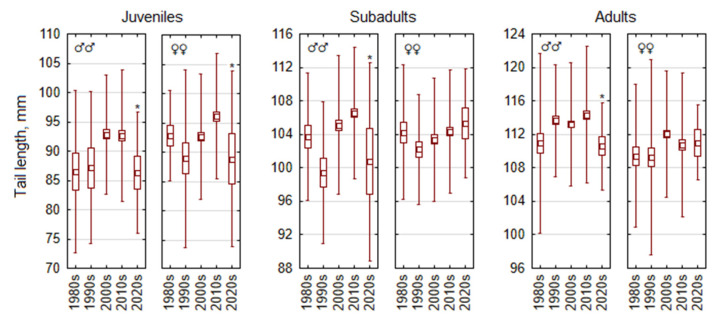
Temporal changes in tail length of *Apodemus flavicollis* from 1980 to 2024 depending on sex and age. Central tendency and variability are represented by mean (central square), standard error (box), and standard deviation (whiskers). Significant trait changes in the 2020s compared with the 2010s are marked with an asterisk.

**Figure 5 life-15-00322-f005:**
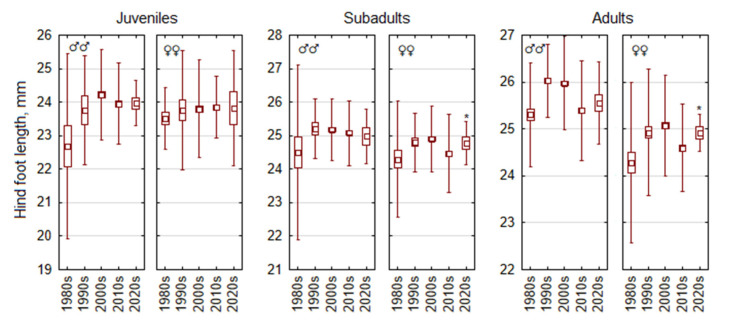
Temporal changes in hind foot length of *Apodemus flavicollis* from 1980 to 2024 depending on sex and age. Central tendency and variability are represented by mean (central square), standard error (box), and standard deviation (whiskers). Significant trait changes in the 2020s compared with the 2010s are marked with an asterisk.

**Figure 6 life-15-00322-f006:**
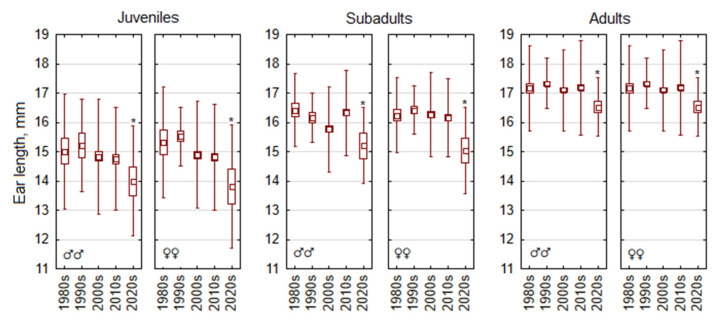
Temporal changes in ear length of *Apodemus flavicollis* from 1980 to 2024 depending on sex and age. Central tendency and variability are represented by mean (central square), standard error (box), and standard deviation (whiskers). Significant trait changes in the 2020s compared with the 2010s are marked with an asterisk.

**Figure 7 life-15-00322-f007:**
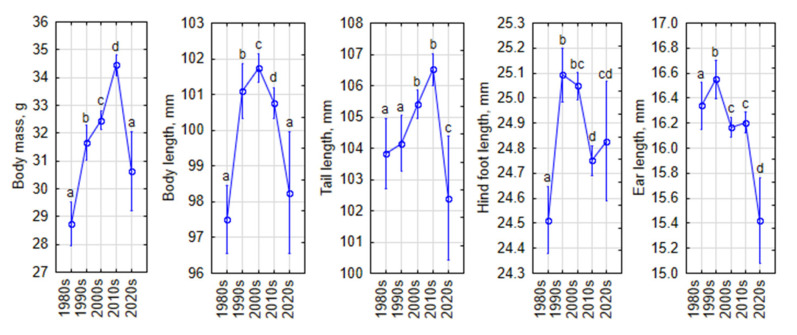
Temporal changes in morphometric traits of *Apodemus flavicollis* from 1980 to 2024, computed for covariates of sex and age at their means. Vertical bars indicate 0.95 confidence intervals. Statistically significant differences are indicated by different letters.

**Table 1 life-15-00322-t001:** Number of *Apodemus flavicollis* captured and dissected by decade, indicating age structure of the sample. ad—adult, sub—subadult, juv—juvenile individuals.

Decade	*Apodemus flavicollis*	
	Captured	Dissected (ad/sub/juv)
1980s	3695	264 (143/74/47)
1990s	2825	584 (382/129/71)
2000s	11,600	2054 (979/575/494)
2010s	7764	1925 (750/619/493)
2020s	3319	839 (373/256/206)

**Table 2 life-15-00322-t002:** Changes in the proportion of *Apodemus flavicollis* in the small mammal community (%) in different habitats by decade. n/a—not examined. Superscripts indicate significance of differences between decades.

Habitat	1980s	1990s	2000s	2010s	2020	Total
Forest	7.9 ^a^	16.8 ^b^	23.8 ^c^	28.4 ^d^	33.6 ^ed^	21.2
Meadow	5.9 ^a^	8.8 ^ab^	9.4 ^b^	7.5 ^ac^	36.2 ^d^	10.6
Wetland	3.7 ^a^	15.0 ^b^	21.9 ^c^	7.3 ^d^	n/a	8.1
Agricultural	20.0 ^ab^	18.9 ^a^	18.3 ^a^	20.7 ^a^	32.7 ^b^	22.8
Overall	6.5 ^a^	17.5 ^b^	18.3 ^b^	22.0 ^c^	28.2 ^d^	18.5

**Table 3 life-15-00322-t003:** Overlap of morphological traits between age groups and sexes in *Apodemus flavicollis*.

Trait	Males			Females		
	Juveniles	Subadults	Adults	Juveniles	Subadults	Adults
Body mass, g	7.3–35.6	17.3–48	18.1–66.5	6.0–35.0	14.5–46.6	15.0–57.0
Body length, mm	57.5–106.2	76.5–118	71.0–129.5	56.5–106.5	74.5–116.0	74.9–127.1
Tail length, mm	58.6–140.2	75.3–126	59.0–137.2	23.7–123.6	78.2–123.8	25.2–132.2
Hind foot length, mm	17.0–27.5	21.8–28	22.0–28.3	14.8–26.5	21.5–27.5	21.2–28.3
Ear length, mm	8.3–19.5	11.5–20	12.1–21.1	8.0–18.9	11.0–20.5	10.5–21.3

## Data Availability

This is ongoing research, therefore data are available from the corresponding author upon request.

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
