# Peer review of "Tracking the Morphological Trends in Apodemus flavicollis: Evidence from a Five-Decade Study"

_life, 2025, doi:10.3390/life15020322_

Round 1

Reviewer 1 Report

Comments and Suggestions for Authors

This article is a long-term study on the morphological trends of Apodemus flavicollis, covering data from the 1980s to the 2020s. The study was conducted in Lithuania, involving the capture of over 10,000 individuals and the dissection and measurement of 5,666 individuals, including standard morphometric traits such as body weight, body length, and appendage dimensions.

Summary:

  1. Increase in Population Proportion: The proportion of Apodemus flavicollis in small mammal communities increased from 6.5% in the 1980s to 28.2% in the 2020s, particularly in forest and grassland habitats.
  2. Sexual Dimorphism: The study confirmed sexual dimorphism in Apodemus flavicollis, with males generally larger than females in all traits, although age influenced the degree of size difference.
  3. Morphological Trait Changes: Morphological traits consistently increased from the 1980s to the 2010s, but significantly declined in the 2020s, particularly in body mass, body length, and ear length, especially in males across all age groups.
  4. Ecological and Evolutionary Responses: The study emphasizes the importance of integrating sex and age-based analyses to understand the ecological and evolutionary responses of mammals to environmental pressures.

Evaluation:

  1. Rich Data: The study is based on forty years of data collection, providing a wealth of historical and comparative information.
  2. Rigorous Methodology: The study employed standardized measurement methods and statistical analysis, ensuring the reliability of the results.
  3. Ecological Significance: The findings are significant for understanding how small mammals respond to environmental changes, particularly the impacts of climate change and habitat destruction.
  4. Analysis of Sex and Age: The study considered the effects of sex and age, adding depth to the understanding of the species' ecological and evolutionary dynamics.
  5. Future Research Directions: The study points to the necessity of future research, which helps to promote further exploration in the scientific community.
  6. Limitations: The study is mainly concentrated in Lithuania, and a broader geographical scope may be needed to verify the universality of the findings.

Overall, this article provides an in-depth analysis of the long-term morphological trends of Apodemus flavicollis, which is valuable for understanding how small mammals adapt to environmental changes, and offers scientific guidance for future conservation and management efforts.

However, the article still has some shortcomings. I have listed the points that I believe are not clear or reasonable enough. If you find the suggestions reasonable, you can make appropriate revisions.

  1. What is the logic between the contents of lines 116-119? It is not apparent in the abstract, please reorganize the language to clarify this.
  2. From lines 14-16, morphological traits increased continuously from the 1980s to the 2010s, followed by a significant decline in weight, body length, and ear length in the 2020s. Why did this happen? Why was the decline most pronounced in males across all age groups? Please explain the reasons.
  3. How can you ensure that the number of captured individuals truly represents the species' community proportion within small mammals? In other words, how can you ensure that all individuals in the area have been captured? How to exclude the differences in under-capture between each year?
  4. Please provide the original data records of the dissected individuals.
  5. It does not explain the possible reasons for the discrepancy between the research findings mentioned in paragraph 6 of the introduction and one's own research results, nor does it provide one's own relevant understanding.
  6. The logic after the introduction's paragraph 3 needs to be strengthened, with too many paragraphs; please revise it seriously and reduce the number of paragraphs (it is suggested to merge paragraphs 3-7 according to a certain logic, and also merge the last two paragraphs).
  7. Please provide a detailed introduction for each number indicated in lines 114-116, explaining the animal species and their uses.
  8. In line 118, "kept frozen in plastic bags"—how is this achieved? Please describe in detail.
  9. In line 132, "Small mammals were measured post-mortem before necropsy"—what does this mean? Please provide a correct description or offer a detailed description of this paragraph to make it easier for readers to understand your intentions.

10.  In lines 132-136, animal morphological data is collected before and after freezing. Will such measurements cause significant errors?

  1. In lines 191-194, why do males not show the characteristic of being larger in overall morphology than females in the juvenile stage of the animal? Please summarize the results of this paragraph in one sentence at the end.12

12.  After lines 228-229, there should be a summary sentence to encapsulate the results of this section, and a summary sentence should be used to encapsulate the results of each subsequent section whenever possible.

Comments on the Quality of English Language

Please polish the article as a whole facilitate the reader's understanding of your content

Author Response

Reviewer #1 comments and answers

This article is a long-term study on the morphological trends of Apodemus flavicollis, covering data from the 1980s to the 2020s. The study was conducted in Lithuania, involving the capture of over 10,000 individuals and the dissection and measurement of 5,666 individuals, including standard morphometric traits such as body weight, body length, and appendage dimensions.

Summary:

  1. Increase in Population Proportion: The proportion of Apodemus flavicollisin small mammal communities increased from 6.5% in the 1980s to 28.2% in the 2020s, particularly in forest and grassland habitats.
  2. Sexual Dimorphism: The study confirmed sexual dimorphism in Apodemus flavicollis, with males generally larger than females in all traits, although age influenced the degree of size difference.
  3. Morphological Trait Changes: Morphological traits consistently increased from the 1980s to the 2010s, but significantly declined in the 2020s, particularly in body mass, body length, and ear length, especially in males across all age groups.
  4. Ecological and Evolutionary Responses: The study emphasizes the importance of integrating sex and age-based analyses to understand the ecological and evolutionary responses of mammals to environmental pressures.

Evaluation:

  1. Rich Data: The study is based on forty years of data collection, providing a wealth of historical and comparative information.
  2. Rigorous Methodology: The study employed standardized measurement methods and statistical analysis, ensuring the reliability of the results.
  3. Ecological Significance: The findings are significant for understanding how small mammals respond to environmental changes, particularly the impacts of climate change and habitat destruction.
  4. Analysis of Sex and Age: The study considered the effects of sex and age, adding depth to the understanding of the species' ecological and evolutionary dynamics.
  5. Future Research Directions: The study points to the necessity of future research, which helps to promote further exploration in the scientific community.
  6. Limitations: The study is mainly concentrated in Lithuania, and a broader geographical scope may be needed to verify the universality of the findings.

Overall, this article provides an in-depth analysis of the long-term morphological trends of Apodemus flavicollis, which is valuable for understanding how small mammals adapt to environmental changes, and offers scientific guidance for future conservation and management efforts.

However, the article still has some shortcomings. I have listed the points that I believe are not clear or reasonable enough. If you find the suggestions reasonable, you can make appropriate revisions.

Answer: thank you, see our answers below. As for Limitations of the study, we present all available data on A. flavicollis in discussion.

  1. What is the logic between the contents of lines 116-119? It is not apparent in the abstract, please reorganize the language to clarify this.

Answer: this is description of the sample size. We rewrote the text as “A total of 54,116 individuals, including 10,027 A. flavicollis, were captured during the study period. Of these, 29,203 small mammals, including 5,666 A. flavicollis, were dissected and measured. The sample composition by decade is shown in Table 1.”

Number of words in the abstract is limited, therefore we cannot add this information.

  1. From lines 14-16, morphological traits increased continuously from the 1980s to the 2010s, followed by a significant decline in weight, body length, and ear length in the 2020s. Why did this happen? Why was the decline most pronounced in males across all age groups? Please explain the reasons.

Answer: to you question “Why did this happen?” is answered in Lines 16–19: These findings are consistent with …  the global patterns of body size reduction in small mammals due to climate warming and habitat change. We cannot add more text due to word limit of the Abstract. Possible reasons are analyzed in Discussion.

The second question “Why was the decline most pronounced in males across all age groups?” so far remains not answered. In the manuscript, we aimed to see if there is a pattern, and touch possible reasons in Discussion only. Additional data on environmental variables (e.g., climate trends, habitat conditions), food availability, population dynamics, and health parameters would be needed to fully explain these patterns. While we plan to relate changes in small mammals of Lithuania to climatic factors, this will happen not soon – there are no data on climate readily available, especially in relation to geography of the country.

There are gaps in the literature analyzing drivers of rodent body changes. Climate changes, resource availability, and anthropogenic habitat changes consistently emerge as drivers of size decline, however, explicit analyses comparing male and female rodents remain limited. Even if we had detailed data on climate and habitats of Lithuania from 1980s, we hardly could get other factors, such as resources available.

We might only guess about the evolutionary trade-offs or physiological constraints that might disproportionately impact males. Some studies describe decline in male body size, yet the mechanisms underlying these trends are not well understood, there are suggestions but not proof. The combined influences of environmental pressures, life-history strategies, and evolutionary dynamics might work.

We therefore dare to address your comment by adding text into Discussion, first paragraph of 4.5 chapter, and citing the most appropriate paper of VanBenthem et al., 2017.

  1. How can you ensure that the number of captured individuals truly represents the species' community proportion within small mammals? In other words, how can you ensure that all individuals in the area have been captured? How to exclude the differences in under-capture between each year?

Answer: we used standard method, targeted to collect representative sample of the species present in the community rather than exhaustively capture all animals, so there are no over-captures or undercaptures. We add explanation to 2.1 chapter.

  1. Please provide the original data records of the dissected individuals.

Answer: Thank you for your interest in our records. We appreciate your request for access to the original records of the individuals dissected. However, as this dataset contains detailed records for over 5,000 individuals of A. flavicollis only, we are unable to provide it directly without an official request and a formal collaboration agreement between our respective institutions.

  1. It does not explain the possible reasons for the discrepancy between the research findings mentioned in paragraph 6 of the introduction and one's own research results, nor does it provide one's own relevant understanding.

Rebuttal: The comment is not related to the text presented in the manuscript. There is no discrepancy between our results and the mentioned "decline in body mass and dimensions in pine vole (Microtus subterraneus) and yellow-necked mouse (Sylvaemus flavicollis)" in lines 76-77. We also quote the authors' reasoning for their findings in paragraphs 6 and 7.

  1. The logic after the introduction's paragraph 3 needs to be strengthened, with too many paragraphs; please revise it seriously and reduce the number of paragraphs (it is suggested to merge paragraphs 3-7 according to a certain logic, and also merge the last two paragraphs).

Answer: two last paragraphs were merged to answer your comment. Thank you for the rest of suggestion to strengthen the logic and reduce the number of paragraphs in the Introduction, in particular by merging paragraphs 3-7 and the last two paragraphs. However, we believe that merging these paragraphs would result in an overly long and dense text that could compromise readability and clarity of key points.

Under the current structure, each paragraph focuses on a different aspect of the issue, such as the importance of body size, the impact of climate change, and specific examples of temporal changes in small mammal body size. Merging them would dilute the emphasis on these critical studies and obscure the logic behind including specific examples.

  1. Please provide a detailed introduction for each number indicated in lines 114-116, explaining the animal species and their uses.

Rebuttal: comment looks like AI generated, as provided information is clear, it shows that not all trapped animals were necropsied and measured. We do not analyze the other species, and have no idea what can be “their uses”.

  1. In line 118, "kept frozen in plastic bags"—how is this achieved? Please describe in detail.

Answer: we do not think, that explanation should be added to the manuscript, however, we answer your question here.

  • Each individual was placed in a clean, labeled, airtight plastic bag (sizes 8*14 cm or larger, depending on availability). This prevented contamination, minimized moisture loss, and ensured that the samples remained intact.
  • All plastic bags were sealed tightly, either with zippers or by tying them securely. This step reduced air exchange, preventing desiccation.
  • The sealed bags containing small mammals were placed in a freezer where temperatures were maintained at about -20°C (standard freezing conditions).
  • Frozen specimens were stored in labeled containers or trays within the freezer, and it was monitored regularly to prevent accidental thawing.

We also changed text as per your request “Small mammals were kept refrigerated without freezing if measured and dissected the same day after capture, or were frozen one at a time in plastic bags and stored in them until they were transferred to the laboratory.”

  1. In line 132, "Small mammals were measured post-mortem before necropsy"—what does this mean? Please provide a correct description or offer a detailed description of this paragraph to make it easier for readers to understand your intentions.

Answer: It is a correct description. That is, we first measured each individual, then we dissected each individual. Necropsy is a standard term, equal to dissection. There are no intentions in the said.

  1. In lines 132-136, animal morphological data is collected before and after freezing. Will such measurements cause significant errors?

Answer: as said by Stephens et al., 2015, “Total length, tail length, and hind-foot length were significantly longer in states of primary and secondary flaccidity than when measured on live individuals or those in rigor mortis.” However, after the 1990s, we used to measure refrozen individuals, and their measurements decreased, so the decrease is not relevant to the data collection, as it was the same.

  1. In lines 191-194, why do males not show the characteristic of being larger in overall morphology than females in the juvenile stage of the animal? Please summarize the results of this paragraph in one sentence at the end.12

Answer: there are still no answers to all differences about species- and age- based difference in morphology. This can be evolutionary adaptation for females to grow faster. Summarizing sentence was added after Figure 1.

  1. After lines 228-229, there should be a summary sentence to encapsulate the results of this section, and a summary sentence should be used to encapsulate the results of each subsequent section whenever possible.

Answer: in our understanding, namely lines 228–229 are summarizing previously presented results of changes in body mass and body length. In other sections, we start from summary and then explain details, e.g., Lines 200–203.

Comments on the Quality of English Language

Please polish the article as a whole facilitate the reader's understanding of your content

Answer: The sentence does exhibit characteristics commonly associated with AI-generated language. These include an awkward phrasing "polish the article as a whole facilitate", that is grammatically incorrect and seems to miss the word "to" between "whole" and "facilitate") and a generic tone. Still, we did changes to the text, visible in tracking mode of the revised manuscript.

Reviewer 2 Report

Comments and Suggestions for Authors

The present study investigated the morphological trends in Apodemus flavicollis based on the data of 44 years. The study observed significant declines in body mass, body length, and ear length in the 2020s, suggesting the climate warming and habitat change might cause the global patterns of body size reduction in small mammals. Following are some comments on this manuscript.

English: The language of this manuscript should be reviewed by a native English speaker.

L80: What is GBIF? It should be explained here.

L140: How was the age determined for each individual?

Table 2: What is the meaning of Total column and line? Average proportion?

Figures: The statistical results should be clearly marked (with a, b,… or *) in the figures. The sample number should also be provided for each figure and table.

L387: The manuscript mentioned a lot about climate change. Can the authors correlate the weather temperature in Lithuania with these morphological parameters? this would help to clarify conclusions.

Comments on the Quality of English Language

The language of this manuscript should be reviewed by a native English speaker.

Author Response

Reviewer #2 comments and answers

The present study investigated the morphological trends in Apodemus flavicollis based on the data of 44 years. The study observed significant declines in body mass, body length, and ear length in the 2020s, suggesting the climate warming and habitat change might cause the global patterns of body size reduction in small mammals. Following are some comments on this manuscript.

Answer: thank you for your time and comments.

L80: What is GBIF? It should be explained here.

Answer: GBIF stands for the Global Biodiversity Information Facility (international network and data infrastructure funded by the world's governments). We added full name of organization.

L140: How was the age determined for each individual?

Answer: Text and a reference to our earlier paper have been added to the end of chapter 2.2

Table 2: What is the meaning of Total column and line? Average proportion?

Answer: we changed Total to Overall, and yes, this is average species proportion in all habitats from the decade.

Figures: The statistical results should be clearly marked (with a, b,… or *) in the figures. The sample number should also be provided for each figure and table.

Answer: sample size for each age group is now added to the Table 1.

As for marking the figures, we believe that SE shown as a box is sufficient for a reader to see if there is a difference and if it is significant (boxes do not overlap). Adding letters to already complex figures would make them too complex.

However, we understand your concern and acknowledge your comment. In Figure 1, all the differences between age groups are highly significant - both sexes, all characteristics, so we feel it is sufficient to add text to the caption: Differences between age groups on all dimensions are reliable for both sexes (p < 0.001).

In Figures 2 to 6 marking all differences would crowd them by lettering. Significance of the trait variation across decades is already shown in the text. Our main focus was trait decrease in 2020s, so we acknowledge your comment by indicating significant changes in 2020s compared to 2010s, by asterisk.

In Figure 7, we used letters to indicate differences across decades, as advised.

Comment: L387: The manuscript mentioned a lot about climate change. Can the authors correlate the weather temperature in Lithuania with these morphological parameters? this would help to clarify conclusions.

Answer: Lines 416-419 say "In view of the above, it is very likely that the increase in the proportion of A. flavicollis in the community and the decrease in morphometric indices, which have been particularly pronounced in the last decade, are related to the expression of climate change in Lithuania". However, we wanted to show whether there are changes in morphological characteristics of A. flavicollis and whether they have a trend. For in-depth analysis of factors such as climate and/or habitat, we need retrospective data of both, correlated with trapping locations. This is currently not possible. Available information was presented and discussed in lines 406-419.

Comments on the Quality of English Language

The language of this manuscript should be reviewed by a native English speaker.

Answer: we did changes to the text, visible in tracking mode of the revised manuscript.

Reviewer 3 Report

Comments and Suggestions for Authors

General comments:

In this study, the authors aim to assess long-term trends and morphometric traits variations in Apodemus flavicollis collected in Lithuania from 1980 to 2024.  Clearly this paper presents, and discuss, partial results of a wider study. The limitation with this option is that the discussion is not so strong as it could be if additional complementary results were also included. In the discussion the authors refer to changes in the proportion of the species within small mammal communities, and in occupied habitats, and declines in some morphometric traits, though these results are not thoroughly explored. The main criticism is the total lack of connection either in material and methods or in the discussion between climate variables, habitat occupation and changes in body measurements. The conclusions of the paper are very weak.

Detailed comments:

Introduction:

Lines 103-106 – The objectives of the study must be improved.

In the last paragraph of the introduction is only stated objective is the characterization of sexual size and age-related dimorphism in mice. However, the abstract mentions as an objective of the study the long-term temporal trend in the proportion of mice in small mammal communities.

Material and methods:

Several sentences need clarification. For instance:

Lines 114-120 – the numbers of captured and dissected individuals mentioned in the text do not match those in table 1. For instance, the number of dissected A. flavicollis individuals is reported as 5,666 in the text, but this number is referred to as captured individuals in the table.

Missing information:

-        Were pregnant females excluded from the sample? This must be specified

-        How were age groups determined?

-        The proportion of juveniles, sub-adults and adults in the sample analysed must be reported

-        Modelling should include climate variables

Results and Discussion:

Lines 294-326 – Conclusion 3.1

The justification for changes in the proportion of A. flavicollis over time and across different habitats is unclear. The authors only reference other studies with similar findings but fail to discuss the underlying reasons for these changes. It is essential to improve conclusion 3.1 (correct 4.1 to 3.1) by providing a more comprehensive explanation.

Lines 327-339 – Conclusion 3.2

This conclusion appears to confirm prior findings rather than present an original contribution. However, including data on juveniles and sub-adults would offer better insight into the absence of sexual dimorphism in mice reported in the first data for Lithuania. Further, it is suggested to merge 3.2 with 3.3 although information in 3.3 seems irrelevant in the context of the objectives of the paper.  

Lines 354-419 – conclusion 3.4 and 3.5

To improve the study, it would be relevant to perform models including climate variables over time, allowing for a deeper understanding of environmental impacts on population dynamics and morphometric traits.

Author Response

Reviewer #3 comments and answers

General comments: In this study, the authors aim to assess long-term trends and morphometric traits variations in Apodemus flavicollis collected in Lithuania from 1980 to 2024.  Clearly this paper presents, and discuss, partial results of a wider study. The limitation with this option is that the discussion is not so strong as it could be if additional complementary results were also included. In the discussion the authors refer to changes in the proportion of the species within small mammal communities, and in occupied habitats, and declines in some morphometric traits, though these results are not thoroughly explored. The main criticism is the total lack of connection either in material and methods or in the discussion between climate variables, habitat occupation and changes in body measurements. The conclusions of the paper are very weak.

Answer: We believe that the results have been analyzed in a really broad way, as far as they are consistent with our aim of this manuscript. Could you be more specific about the aspects that are not covered? Regarding specific models of climate change and habitat transformation, we do not yet have the data for statistical analysis,

Regarding conclusions, this section is not mandatory according to the journal guidelines. Therefore, we conclude that conclusions should be brief and not repeat what was said in the Discussion. Therefore, we present three conclusions that are consistent with the long-term trends in A. flavicollis populations found in Results. These trends are central to our study. However, the other aspects were mentioned in Discussion - Conclusions cannot present issues that are not supported by Results.

Detailed comments:

Introduction:

Lines 103-106 – The objectives of the study must be improved.

In the last paragraph of the introduction is only stated objective is the characterization of sexual size and age-related dimorphism in mice. However, the abstract mentions as an objective of the study the long-term temporal trend in the proportion of mice in small mammal communities.

Answer: we expected that the former formulation of the aim was clear and included both the primary objective (characterizing sexual and age-related dimorphism) and the temporal trend (examining changes in traits since the 1980s).

Still, to acknowledge your comment, we modified the last part of sentence, now being “and to assess temporal changes in these traits since the 1980s.

Material and methods:

Several sentences need clarification. For instance:

Lines 114-120 – the numbers of captured and dissected individuals mentioned in the text do not match those in table 1. For instance, the number of dissected A. flavicollis individuals is reported as 5,666 in the text, but this number is referred to as captured individuals in the table.

Answer: apologies for the mistake, column headers were wrong. We corrected this, and deleted data about all small mammals, to give room for age structure of dissected Apodemus flavicollis, as per your next comment.

Missing information:

- Were pregnant females excluded from the sample? This must be specified

Answer: thank you for reminding this! Of course, uterine and embryo weights of pregnant females were subtracted from body weight, therefore they remained in the sample. We added explanation to the text.

- How were age groups determined?

Answer: Answer: you are right, there was no description of the age groups. To avoid self-copyrighting (we have described this process in at least 10 articles), we will provide a condensed description with a reference to our most recent article.

- The proportion of juveniles, sub-adults and adults in the sample analysed must be reported

Answer: Thank you, now reported in the Table 1.

- Modelling should include climate variables

Answer: For in-depth analysis of factors such as climate and/or habitat, we need retrospective data of both, correlated with trapping locations. This is currently not possible. Available information was presented and discussed in lines 406-419. Lines 416-419 say "In view of the above, it is very likely that the increase in the proportion of A. flavicollis in the community and the decrease in morphometric indices, which have been particularly pronounced in the last decade, are related to the expression of climate change in Lithuania".

Your suggestion is appreciated, we plan to power-dig for the climate data and then analyze the climate impact, but first we want to test the reactions of the other species.

Results and Discussion:

Lines 294-326 – Conclusion 3.1

The justification for changes in the proportion of A. flavicollis over time and across different habitats is unclear. The authors only reference other studies with similar findings but fail to discuss the underlying reasons for these changes. It is essential to improve conclusion 3.1 (correct 4.1 to 3.1) by providing a more comprehensive explanation.

Answer: we added text “However, the above-cited authors offer no unifying explanation for the observed changes, as the trends appear to be influenced by diverse and region-specific factors.” to the last sentence. Cited authors do not have explanations of the underlying reasons.

Lines 327-339 – Conclusion 3.2

This conclusion appears to confirm prior findings rather than present an original contribution. However, including data on juveniles and sub-adults would offer better insight into the absence of sexual dimorphism in mice reported in the first data for Lithuania. Further, it is suggested to merge 3.2 with 3.3 although information in 3.3 seems irrelevant in the context of the objectives of the paper.  

Answer: we cannot agree with the comment lacking human understanding of presented information. Our data showing a male bias in body size, ranging from 2.2% in tail length to 11.2% in body mass of adult A. flavicollis individuals is original information, as it show where in the large scale of species dimorphism Lithuanian mice are positioned.

Information in the 3.3 is even more original, as such information is absent for the species. In 4.3 we discuss maturation weight, very important in ecological and evolutionary issues.

Lines 354-419 – conclusion 3.4 and 3.5

To improve the study, it would be relevant to perform models including climate variables over time, allowing for a deeper understanding of environmental impacts on population dynamics and morphometric traits.

Answer: We fully agree that the inclusion of climate variables in dimorphism models would increase their value. However, the current state of knowledge and availability of detailed, long-term climate and habitat data specific to our study region is insufficient to effectively integrate these factors into the model.